

# Flowering-associated gene expression and metabolic characteristics in adzuki bean (*Vigna angularis L.*) with different short-day induction periods

Weixin Dong[1,2], Dongxiao Li[2], Lei Zhang[2,3], Peijun Tao[2] and Yuechen Zhang[2]

[1] College of Agronomy and Medical, Hebei Open University, Shijiazhuang, Hebei, China
[2] College of Agronomy, Hebei Agricultural University, Baoding, Hebei, China
[3] College of Life Sciences, Zaozhuang University, Zaozhuang, Shandong, China

## ABSTRACT

**Background:** The adzuki bean is a typical short-day plant and an important grain crop that is widely used due to its high nutritional and medicinal value. The adzuki bean flowering time is affected by multiple environmental factors, particularly the photoperiod. Adjusting the day length can induce flower synchronization in adzuki bean and accelerate the breeding process. In this study, we used RNA sequencing analysis to determine the effects of different day lengths on gene expression and metabolic characteristics related to adzuki bean flowering time.

**Methods:** 'Tangshan hong xiao dou' was used as the experimental material in this study and field experiments were conducted in 2022 using a randomized block design with three treatments: short-day induction periods of 5 d (SD-5d), 10 d (SD-10d), and 15 d (SD-15d).

**Results:** A total of 5,939 differentially expressed genes (DEGs) were identified, of which 38.09% were up-regulated and 23.81% were down-regulated. Gene ontology enrichment analysis was performed on the target genes to identify common functions related to photosystems I and II. Kyoto Encyclopedia of Genes and Genomes enrichment analysis identified two pathways involved in the antenna protein and circadian rhythm. Furthermore, florescence was promoted by down-regulating genes in the circadian rhythm pathway through the blue light metabolic pathway; whereas, antenna proteins promoted flowering by enhancing the reception of light signals and accelerating electron transport. In these two metabolic pathways, the number of DEGs was the greatest between the SD-5d VS SD-15d groups. Real-time reverse transcription–quantitative polymerase chain reaction analysis results of eight DEGs were consistent with the sequencing results. Thus, the sequencing results were accurate and reliable and eight genes were identified as candidates for the regulation of short-day induction at the adzuki bean seedling stage.

**Conclusions:** Short-day induction was able to down-regulate the expression of genes related to flowering according to the circadian rhythm and up-regulate the expression of certain genes in the antenna protein pathway. The results provide a theoretical reference for the molecular mechanism of short-day induction and multi-level information for future functional studies to verify the key genes regulating adzuki bean flowering.

Corresponding author
Yuechen Zhang,
Zhangyc1964@126.com

# INTRODUCTION

The adzuki bean originated in China and is an important grain crop. As a homologous medicinal food species, it plays an important role in the human diet, with an annual export volume of approximately 40,500 tons (*Clemente & Jimenez-Lopez, 2020*). However, this species undergoes major shedding of flowers and pods, resulting in a low and unstable bean yield (*Gao et al., 2021*). Many recent studies have found that the photoperiod can be set at the seedling stage to regulate the flowering time, resulting in significant improvements in bean yield and quality in the later stage (*Dong et al., 2018*; *Ding et al., 2019*; *Tribhuvan et al., 2020*). There have been few reports on the molecular mechanisms of photoperiodical induction and regulation of legume flowering (*Vanhala et al., 2016*; *Cheng et al., 2023*; *Yang et al., 2021b*). RNA sequencing (RNA-seq) technology can comprehensively identify transcripts, conduct transcriptome analyses, and rapidly identify expressed genes with high sensitivity and accuracy (*Niedziela et al., 2022*). Therefore, using this technology to identify the regulatory pathways and candidate genes involved in plant flowering is important for understanding the molecular mechanisms underlying the process.

As an important phenological stage, the flowering phase initiates plant reproduction and involves complex mechanisms of the transduction of environmental signals, among which the photoperiod is one of the most important environmental factors affecting the flowering time (*Zhao et al., 2018*; *Mahmood et al., 2023*). The molecular mechanisms of flowering in response to the photoperiod have been extensively studied in *Arabidopsis thaliana* (*Takagi, Hempton & Imaizumi, 2023*; *Serrano-Bueno et al., 2020*). As a typical long-day plant, the regulatory mechanism of flowering in *A. thaliana* is related to increased constans (CO) protein levels in the long term and increased flowering locus T (*FT*) mRNA levels (*Kinmonth-Schultz et al., 2021*). Finally, the FT protein is transported to the meristematic tissue through the vascular bundle, promoting differentiation of the floral primordia into floral tissue (*Corbesier et al., 2007*). Under short-day conditions, the COP1/SPA1 complex is formed to degrade the CO protein, thus down-regulating *FT* expression and inhibiting *A. thaliana* flowering (*Kreiss et al., 2023*). Flowering genes in soybean have been studied extensively and 10 quantitative trait loci (namely *E1–E9* and *EJ*) related to flowering time have been identified (*Lin et al., 2021*). Among these loci, *E1–E4*, *E7*, and *E8* regulate the flowering time of soybean through different photoperiods, with *E1* contributing the most to soybean flowering (*Tsubokura et al., 2014*). *E5*, which does not exist separately, may be a result of accidental outcrossing with pollen containing the *E2* allele (*Dissanayaka et al., 2016*). *E9* is a leaky allele of *FT2a* and its abundance is directly related to changes in the soybean flowering time (*Kong et al., 2014*). In addition to the *E* genes, the *FT* gene also plays an important role in the soybean flowering process (*Zhao et al., 2016*). In adzuki bean, three metabolic pathways related to flowering, including plant hormone, circadian rhythm, and antenna protein pathways, have been identified by

comparing the transcriptome sequencing results of plants with different short-day induction periods with their respective controls (*Dong et al., 2022*). The homologues of 13 verified genes related to flowering were identified, indicating that these genes are key candidates for regulating adzuki bean flowering (*Dong et al., 2022*). Other studies have used transcriptome analysis to identify *ELF3, PRR5, PRR7, LHY*, timing of CAB expression 1 (*TOC1*), gigantea (*GI*), *CO*, and *FT* as significantly differentially expressed genes (DEGs) related to flowering after short-day induction in *Phaseolus vulgaris L.* The function of these DEGs is related to biological clock regulation (*Yang et al., 2021b*). In addition, *TOC1* plays a key role in controlling the photoperiodic flowering response through the clock function, up-regulation or down-regulation of *TOC1* leads to changes in downstream genes that regulate the flowering time in other plant species (*Niwa et al., 2007*; *Ma et al., 2020*).

There has been an increase in transcriptome research on plant flowering, and the metabolic pathways and key genes that regulate flowering through the photoperiod in *A. thaliana*, the adzuki bean, and *P. vulgaris* have been described in detail. Additionally, transcriptomics has increasingly been used to study regulation of flowering and flower development in diverse plant species, including lute (*An et al., 2021*), tobacco (*Guan et al., 2021*), lily (*Zhao et al., 2021*), cucumber (*Cai et al., 2020*), and hemp (*Li et al., 2021*). However, the current understanding of the effect of photoperiod on plant flowering needs to be broadened through transcriptome analysis. To the best of our knowledge, transcript information and gene expression profiles of adzuki bean under different short-day induction periods have not yet been determined. Therefore, applying RNA-seq to identify candidate genes related to flowering would help to elucidate the regulatory mechanism of short-day-induced flowering in adzuki bean. In this study, three short-day induction periods of 5 d (SD-5d), 10 d (SD-10d), and 15 d (SD-15d) were implemented, the transcriptome of adzuki bean leaves was analyzed, and differentially expressed transcripts were determined to identify candidate transcripts involved in regulation of flowering. The aim of this study was to understand the early flowering process under different short-day induction periods through gene expression regulation to provide a reference for accelerated breeding of adzuki bean and similar short-day photoperiod crops.

# METHODS

## Experimental material

Portions of this text were previously published as part of a preprint (https://doi.org/10.21203/rs.3.rs-3362672/v1). A late-maturity variety 'Tangshan hong xiao dou,' which is sensitive to short days, was selected as the experimental material. Plants were provided by the Adzuki Bean Breeding Research Group of the Institute of Grain and Oil Crops, Hebei Academy of Agricultural and Forestry Sciences (China). The experiment was conducted in the Teaching and Experimental Base of Hebei Agricultural University, Baoding (longitude: 115°47′, latitude: 38°87′) in 2022. As a typical short-day crop, adzuki bean is highly sensitive to short-day induction; thus, the flowering time and maturity stage are significantly earlier than those under a natural photoperiod.

## Soil fertility

The experimental field was comprised of loam soil, and 400 g of compound fertilizer (N: $P_2O_5$:$K_2O$ = 24:4:8) was applied to each plot (5-m long, 1-m wide) prior to sowing. The nutrient content of the experimental plot was measured after ploughing. The fertility of the cultivated soil layer (0–20 cm) of the experimental plot is shown in Table 1.

## Environmental characteristics of the plot after shading

The environmental characteristics of the plot after shading treatment are shown in Table 2. Compared with the atmospheric environment, the light intensity of the plot after shading was near zero, the relative humidity was significantly increased by 21.41%, and the $CO_2$ concentration and temperature were slightly, but not significantly increased.

## Short-day treatment

Compound fertilizer (400 g) was applied to the experimental plot, followed by ploughing, and then sowing which was carried out on June 24, 2022. Planting was performed in two rows, plant spacing and row spacing were 15 × 40 cm. Plants were grown under natural light until the true leaf unfolded, after which they were placed on a stainless-steel shelf (6-m long, 1.5-m wide, 1-m high) on the upper side of the plots and covered with an opaque cloth. The opaque cloth (10-m long, 5-m wide) comprised two layers of black cloth and two layers red cloth stitched together. The shading treatment involved 10 h of light and 14 h of dark with shading for 5, 10, or 15 d. By adopting a randomized block design, this experiment involved initiating shading treatment with the opaque cloth at 18:00 every day until 08:00 the following morning. After the shading treatment, all plants were grown to maturity under natural light. At the initial flowering and pod-setting stages, 0.4% potassium dihydrogen phosphate was sprayed and irrigation. There were three replicates for each shading treatment (SD-5d-1, SD-5d-2, and SD-5d-3; SD-10d-1, SD-10d-2, and SD-10d-3; SD-15d-1, SD-15d-2, and SD-15d-3), with a total of nine plots. After 5, 10, and 15 d of shading, the middle leaf samples of the top trifoliate leaves were uniformly removed at 09:00 from 5–6 plants in each group. After sampling, the leaves were immediately frozen in liquid nitrogen and stored at −80 °C. RNA extraction and sequencing were performed 1 week later.

## Meteorological factor determination

Light intensity was measured 20–30 cm above the canopy of the adzuki bean community using a TES1332 illuminometer (provided by College of Plant Protection, Agricultural University of Hebei, TES1332; TES, Taipei, Taiwan). The $CO_2$ concentration was measured using an Li-6400 portable photosynthetic meter (LI-COR, Lincoln, NE, USA). Temperature and humidity were measured using a HOBO Pro V2 series data logger (U23-002; Onset Computer Corporation, Bourne, MA, USA), which automatically counted and recorded data every hour.

**Table 1 Determination of soil nutrients content in experiment field.**

| Experimental site | Year | Determination index | | | | |
|---|---|---|---|---|---|---|
| | | Organic matter (%) | Total nitrogen (%) | Available nitrogen (ppm) | Available phosphorus (ppm) | Available potassium (ppm) |
| Teaching and experimental base of Hebei Agricultural University | 2022 | 1.61 | 0.0969 | 85.88 | 66.761 | 189.2 |

**Table 2 Changes in field microclimate with shading treatment.**

| Treatment | Illumination intensity (lux) | CO$_2$ concentration (ppm) | Relative humidity (%) | Temperature (°C) |
|---|---|---|---|---|
| Ambient environment | 58,865.41 ± 1,547.23a | 478.46 ± 15.56a | 75.35 ± 5.83b | 25.88 ± 1.33a |
| Environment of plots | 11.67 ± 4.21b | 534.54 ± 18.76a | 95.88 ± 6.43a | 26.87 ± 1.47a |

**Note:**
Values are means ± S.E, The different letters in each column indicate significant differences, assessed by ANOVA ($P \leq 0.05$).

## Growth index determination

Plant height, stem diameter, and leaf area were measured in three representative plants selected from each plot at the flowering, pod-setting, and grain-filling stages. Plant height was calculated as the distance from the true leaf to the growing point of the plant. The plant stem diameter at the true leaf was measured using a vernier caliper and calculated as: circumference = 2 * 3.14 * (diameter/2). Leaf area was measured using a YMJ-B leaf area measuring instrument (Hangzhou Huier Instrument Equipment Co., Ltd., Hangzhou, China).

## Flowering characteristics

At the flowering stage, three representative plants were selected from each treatment. The advanced flowering days were recorded and the flowering promotion rate was calculated using the following formulae:

Advanced flowering days = number of days from emergence to flowering of plants in the control group – number of days from emergence to flowering of plants in the treatment group.

Flowering promoting rate (%) = ([number of days from emergence to flowering in the control group – number of days from emergence to flowering in the treatment group] × 100)/number of days from emergence to flowering in the control group.

Apical flower buds were removed after short-day induction. After sampling, the buds were fixed, dehydrated and made transparent, paraffin-embedded, and sectioned. The paraffin-embedded sections were stained with hematoxylin and eosin and then observed under a microscope (BS500/BS500-TR biological microscope). Flower bud differentiation in adzuki bean was divided into the pre-differentiation (PD), inflorescence primordium differentiation (IP), flower primordium differentiation (FP), sepal primordium differentiation (SP), petal primordium differentiation (PP), stamen and

carpel primordium differentiation (SCP), and stamen and carpel structural differentiation (SCS) stages according to the division method described by *Jin, Chen & Yu (1995)*.

## RNA extraction and cDNA library construction

Total RNA was extracted using a mirVana miRNA Isolation Kit (Ambion, Austin, TX, USA) following the manufacturer's protocol and DNA was digested using DNase. Eukaryotic mRNA was enriched using oligo coupled to magnetic beads, and the mRNA was fragmented by addition of interrupting reagents. The fragmented mRNA was used as the template for first-strand cDNA synthesis using random hexamers. A second-strand synthesis reaction system was used to synthesize second-strand cDNA, which was further purified using the kit. End repair was then performed, followed by A-tailing and sequencing adapter ligation, fragment size selection, polymerase chain reaction (PCR) amplification, and library construction. After qualification using an Agilent 2100 Bioanalyzer (Agilent, Santa Clara, CA, USA), the constructed libraries were sequenced using a HiSeq2500 or HiSeq X Ten sequencer (Illumina, San Diego, CA, USA) to produce 125 or 150 bp paired-end reads, respectively.

## Screening of DEGs

The fragments per kilobase of transcript per million mapped reads (FPKM) metric (*Roberts et al., 2011*), bowtie2 (*Langmead & Salzberg, 2012*), and cufflinks software (*Roberts & Pachter, 2013*) were used to analyze the transcript levels. Using the *Vigna angularis* reference genome (https://ftp.ncbi.nlm.nih.gov/genomes/all/GCF/001/190/045/) (*Kang et al., 2015*), the number of transcript (protein-coding) reads was obtained for all samples using cufflinks software. Genes with an average number of reads >2 were screened, and related data were standardized using the estimateSizeFactors function of the R package DESeq (*Livak & Schmittgen, 2001*). $P$-values and fold-change values of difference comparisons were calculated using the nbinomTest function. DEGs were identified based on a fold change ≥2 and a false discovery rate <0.05.

## Functional annotation analysis of DEGs

Gene ontology (GO) annotations and functional classification of DEGs ($P < 0.05$ and fold change >2) were performed using Blast2 GO and WEGO. BLAST was used to align gene sequences to those in the Kyoto Encyclopedia of Genes and Genomes (KEGG) database for annotation of biochemical pathways and identification of the regulatory-metabolic network.

## Quantitative reverse transcription–PCR validation

Under the same treatment, other adzuki bean samples were used for quantitative reverse transcription–PCR (qRT–PCR) analysis of eight selected DEGs to verify the accuracy of the sequencing results. RNA was extracted using an RNA extraction kit followed by reverse transcription to produce cDNA (HiScript II Q RT SuperMix for qPCR). Primer 5.0 was used to design the primers; the primer sequences are shown in Table 3. A QuantiFast® SYBR® Green PCR Kit was used to prepare the qRT–PCR reaction system and the transcripts were detected on a fluorescence quantitative PCR instrument (CFX96; Harbin

**Table 3 Primer design for qRT-PCR validation of eight differentially significant genes.**

| Order number | Gene ID | Forward primer sequence (5′→3′) | Reverse primer sequence (5′→3′) |
|---|---|---|---|
| 1 | ACTIN | CTAAGGCTAATCGTGAGAA | CGTAAATAGGAACCGTGT |
| 2 | LOC108331766 | AAAAGGGAGGACCAAGAGCAC | TGAGTGGCACAACACCTGAAT |
| 3 | LOC108322606 | AAGCAAACAAGGAAGGGAAAG | TGAAACATGGCTGCAAAAGAT |
| 4 | LOC108345872 | GAGGCTCCTTCTTACCTGACG | AGTTCACGGTTTCGAGCAAAT |
| 5 | LOC108328079 | AAGAAACGCAGAACTTGACCC | GCTTGAATGGCAAAGATGAGG |
| 6 | LOC108344684 | GAGGCTCCTTCTTACCTGACG | AGTTCAAGGTTCCGAGCAAAT |
| 7 | LOC108335068 | GAGGTGACCGACCCAATTTAC | CACAATCGCCTGAACAAAGAA |
| 8 | LOC108333950 | GGGTTCTTTGTTCAAGCCATT | ACCCAAGCATTGTTAGCCACT |
| 9 | LOC108338432 | CAGGTTGTGCTTATGGGGTTT | AGCGACCATTCTTGAGTTCCT |

Deyuan Science and Technology Development Co., LTD, Harbin, China). The qRT-PCR cycling conditions were as follows: predenaturation at 95 °C for 10 min, followed by 40 cycles of denaturation at 95 °C for 10 s, and annealing/extension at 60 °C for 30 s. Gene expression levels were calculated using the $2^{-\Delta\Delta Ct}$ method (*Lin & Pang, 2019*).

## Data analysis

Raw data were filtered using NGS QC Toolkit software to obtain high-quality, clean reads. FPKM values were calculated using cufflinks software. All data were analyzed using analysis of variance, with three replicates, using Excel 2023 and SPSS 17.0 (SPSS Inc., Chicago, IL, USA). Duncan's new multiple range test (at the 5% probability level) was used to test the differences among the mean values. Images were processed using BS500/BS500-TR biomicroscopy and Photoshop CS6.

# RESULTS

## Growth and flowering characteristics

Different short-day inducement periods had very different effects on apical flower bud differentiation and morphology, the number of days until early flowering, and the flowering promotion rate in adzuki bean (Fig. 1). Plant morphology differed under different short-day induction periods, with a longer short-day induction period resulting in a greater inhibitory effect on plant height of adzuki bean (Fig. 1B). We further observed apical flower bud differentiation and found that the apical flower bud was in the PD stage under the SD-5d treatment; whereas, it was in the SP stage under the SD-10d and SD-15d treatments (Fig. 1A). Plant height, stem diameter, and leaf area decreased with prolongation of short-day induction, and plant height showed significant differences among the three treatments at the flowering, podding and seed-filling stages (Fig. 2A). The stem diameter significantly decreased by 10.91% and 13.27% under the SD-15d treatment compared to that under the SD-5d treatment at the flowering and seed-filling stages, respectively; whereas, there was no significant difference in stem diameter between the SD-5d and SD-10d treatments. There were significant differences in stem diameter

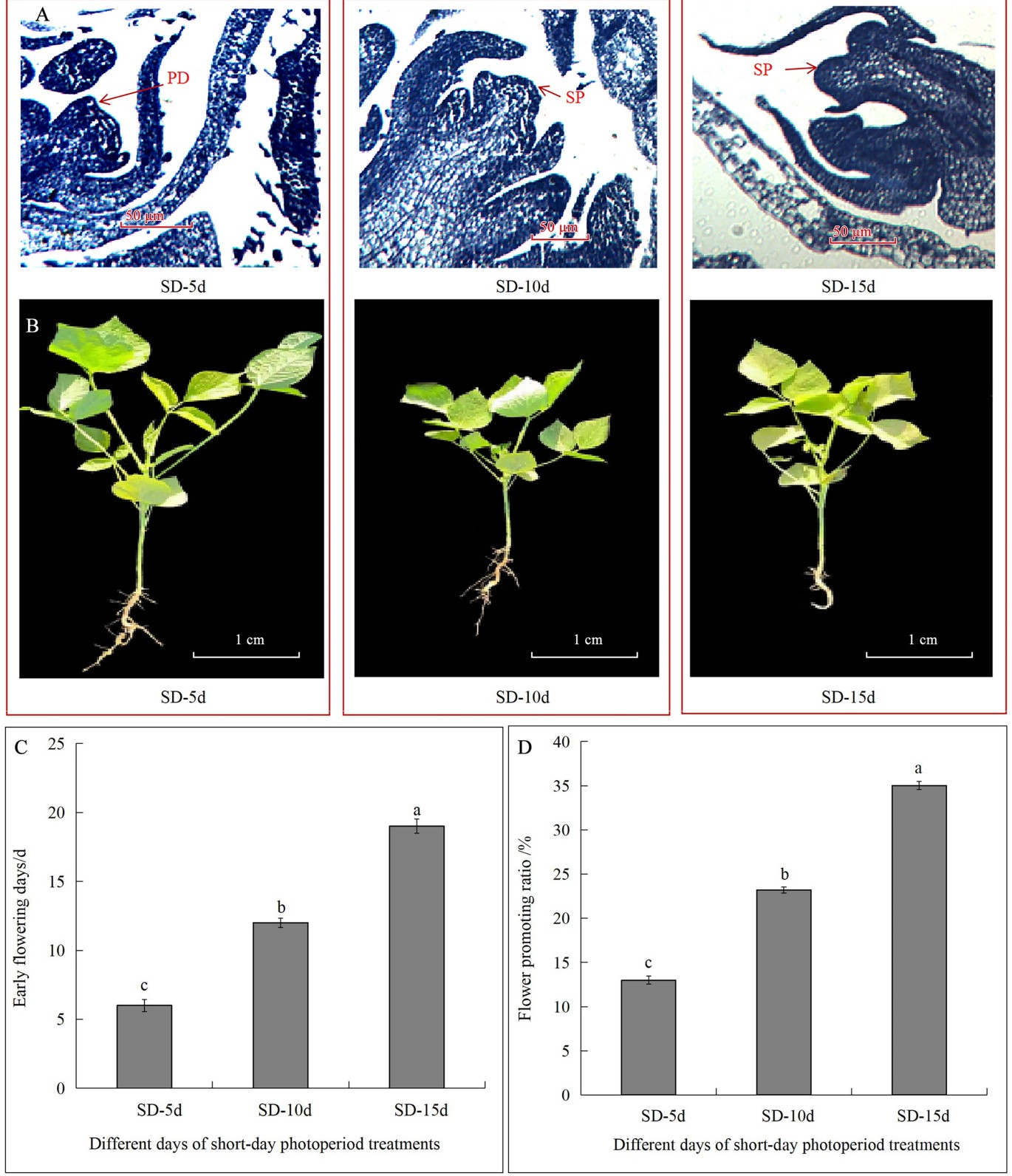

**Figure 1 Effect of short-day induction on apical flower buds differentiate, morphology, early flowering days and flowering promotion rate in adzuki bean.** (A) Apical flower bud was in pre-differentiation stage (PD) under SD-5d treatment, and was in sepal primordium differentiation stage
**Figure 1 (continued)**
(SP) under SD-10d and SD-15d treatments. (B) Plant morphology under different short-day induction. (C) Advanced flowering days of adzuki bean in three different treatment groups. (D) Flowering promoting rate of adzuki bean in three different treatment groups. Note: Values are means ± S.E, The different small letters in each bar indicate significant differences among three treatments, assessed by ANOVA ($P \leq 0.05$).

among the three treatments during the podding stage (Fig. 2B). Significant differences in leaf area were observed among the three treatments at all growth periods (Fig. 2C).

The flowering date was 12 and 7 d earlier in the SD-15d group than in the SD-5d and SD-10d groups, respectively (Fig. 1C), and the SD-15d group improved flowering promotion rates by 22.03% and 11.83% compared to the SD-5d and SD-10d groups, respectively (Fig. 1D). Advanced flowering occurred 6 d later in the SD-10d group than in the SD-5d group, and the flowering promotion rate was 10.20%. These results indicated that short-day induction promoted early flowering of adzuki bean by inhibiting vegetative growth. Short-day induction had a cumulative effect, as a longer short-day inducement resulted in an earlier flowering time and a higher flowering promotion rate.

## Functional analysis of DEGs

Principal component analysis of the samples showed that the dispersion degree among individual samples was significant, and the first principal component (PC1) accounted for 62.71% of the variance. This separation underscored the reproducibility, accuracy, and overall reliability of the data (Fig. 3A). By comparing the data among the three groups using a volcano map, the total number of DEGs was 5,939, including 3,107 up-regulated genes and 2,832 down-regulated genes (Figs. 3B–3D). When comparing SD-5d VS SD-10d, 2,044 DEGs were detected, including 961 up-regulated genes and 1,083 down-regulated genes (Fig. 3B). When comparing SD-5d VS SD-15d, 3,068 genes were detected, including 1,711 up-regulated genes and 1,357 down-regulated genes (Fig. 3C). When comparing SD-10d VS SD-15d, 827 genes were detected, including 435 up-regulated genes and 392 down-regulated genes (Fig. 3D). In addition, by comparing the three groups, 105 common genes were detected, of which 40 were up-regulated and 25 were down-regulated (Figs. 3E–3G). In addition, the SD-5d VS SD-15d comparison showed the highest number of DEGs, indicating that a longer two short-day inducement treatment resulted in the detection of a higher number of DEGs.

Cluster analysis showed that the three treatments significantly affected gene expression (Fig. 3H). Among them, 40 genes were gradually up-regulated with extension of the short-day treatment time; whereas, 25 genes were gradually down-regulated. An additional 40 genes did not show obvious expression patterns, with 25 genes showing up-down-up regulation and 15 genes showing down-up-down regulation in all treatment groups. These results showed that 40 up-regulated and 25 down-regulated genes played key roles in adzuki bean growth and development; however, the other genes had no obvious roles, indicating that the function of different genes varied in response to different short-day induction times.

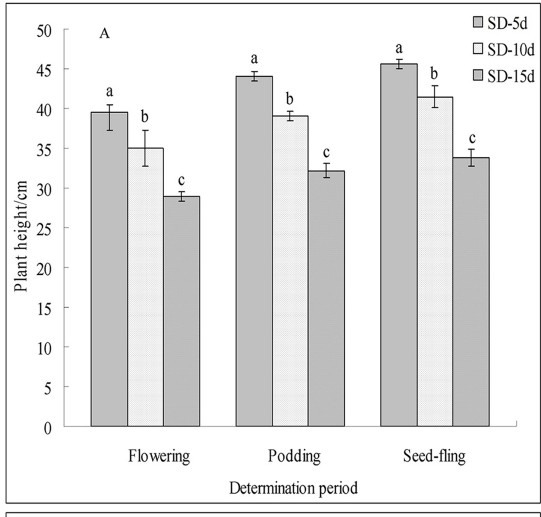

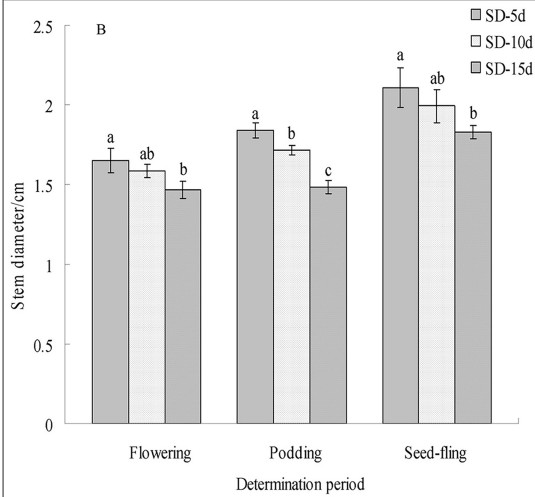

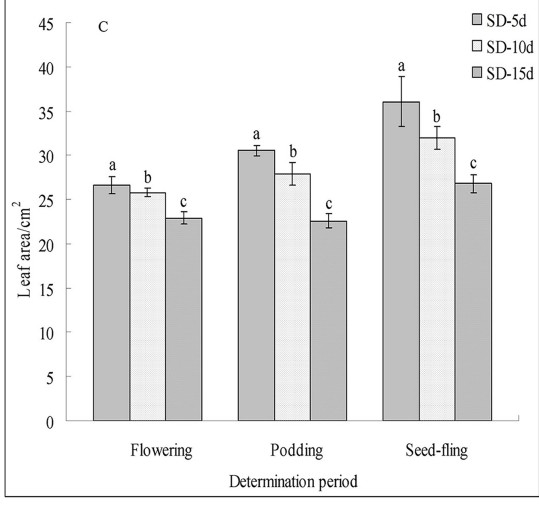

**Figure 2 Plant height, stem diameter and leaf area of adzuki bean under different short-dayinducement times.** Statistics of plant height (A), stem diameter (B) and leaf area (C) under SD-5d, SD-10d, SD-15d treatments. Note: Values are means ± S.E, The different lowercase letters in each bar indicate significant differences among three treatments, assessed by ANOVA ($P \leq 0.05$).

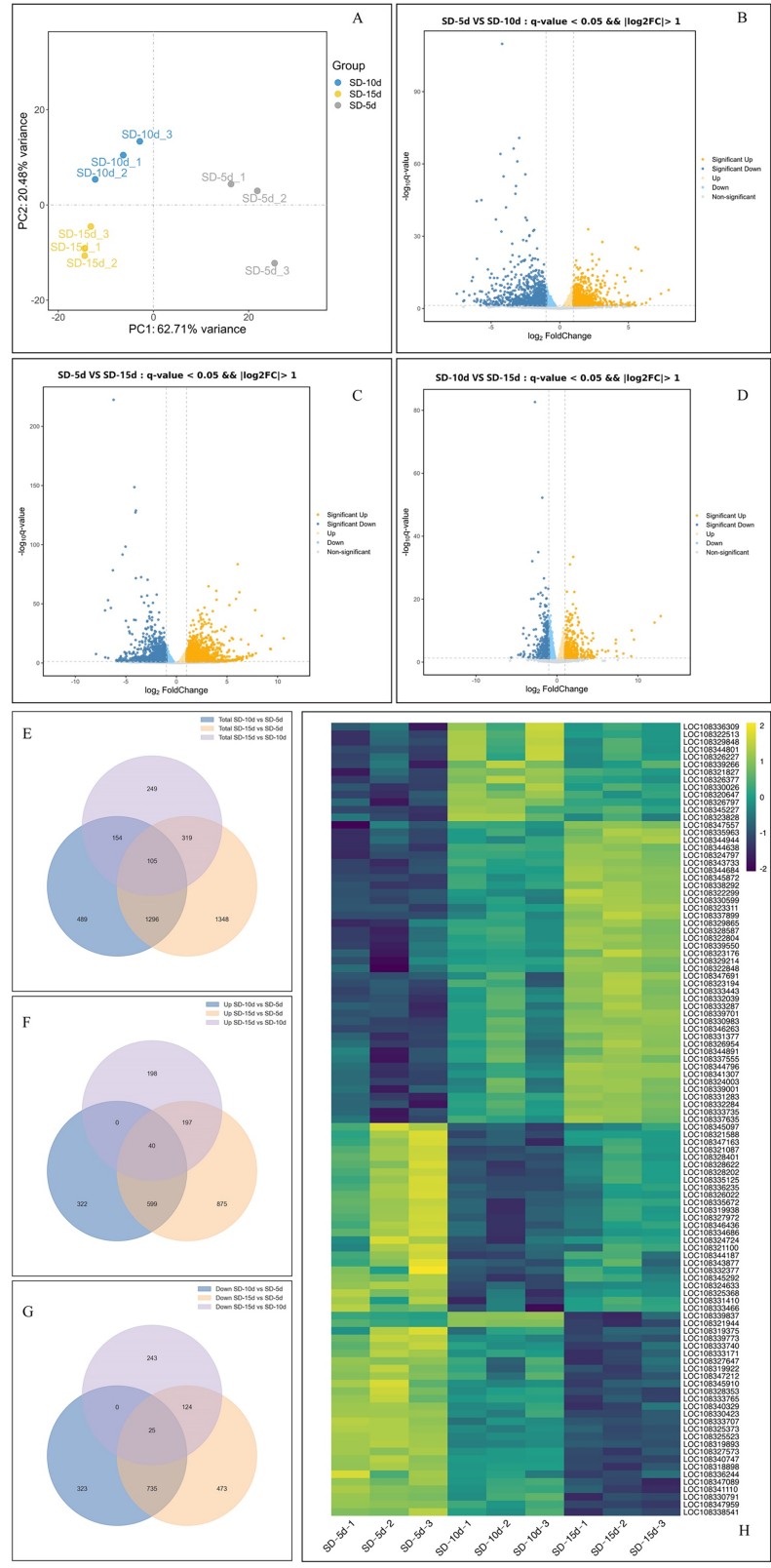

**Figure 3 Analysis of effect on different genes number under different short-day induction.** (A) PCA analysis under SD-5d, SD-10d and SD-15d treatments. Volcano plot under SD-5d VS SD-10d (B), SD-5d VS SD-15d (C) and SD-10d VS SD-15d treatments (D). (E–G) Venn diagram analysis of significantly
**Figure 3** (continued)
regulated genes under different short-day induction treatments. (H) Heatmap clustering of global pattern of the strongly regulated genes conducted using Hierarchical Clustering (HCL) algorithm under different short-day induction treatments. The color scale represents the values of lg FPKM (FPKM-Fragments Per Kilobase of transcript per Million fragments mapped).

## GO functional enrichment analysis

GO terms (DEGs > 2) were screened separately for biological processes, cell components, and molecular functions. GO functional enrichment analysis showed that light-related functions were found among DEGs from all three comparison groups. In the SD-5d VS SD-10d comparison, five up-regulated genes were related to photosystem I (PSI) and II (PSII). In addition, there was a circadian rhythm function related to light among the biological processes, which contained 14 genes, of which six were up-regulated and eight were down-regulated (Fig. 4A). The DEGs in the two comparison groups of SD-5d VS SD-15d and SD-10d VS SD-15d had light-driven functions in PSI and PSII; however, the number of genes involved differed. For the SD-5d VS SD-15d comparison, there were nine and 12 genes associated with PSI and PSII, respectively; the nine genes associated with PSI were all up-regulated, 11 of the 12 genes associated with PSII were up-regulated, and the remaining one gene was down-regulated (Fig. 4B). Both PSI and PSII functions contained five up-regulated genes in the SD-10d VS SD-15d comparison (Fig. 4C). These results suggested that the DEGS in the three comparison groups contained PSI and PSII functions, which are related to light (Fig. 4D), and that photosystem genes in leaf cells may have been activated after different short-day induction times. These DEGs were key regulators of adzuki bean seedling growth and development and were closely related to photoelectron transport under short-day-induced conditions.

## KEGG pathway enrichment analysis of DEGs

KEGG pathway enrichment analysis of DEGs in the three groups was performed to select the top 20 signaling pathways with significant differences in each comparison group. Figure 5 shows that the DEGs identified in the SD-5d VS SD-10d and SD10d VS SD-15d comparisons were enriched in the antenna protein pathway and circadian rhythm pathway two light-related pathways, with different numbers of genes involved in these pathways. The antenna protein pathway in both comparison groups contained five genes that were all up-regulated. In the SD-5d VS SD-10d comparison, of the 16 DEGs in the circadian rhythm pathway, three were up-regulated and 13 were down-regulated (Fig. 5A). For the SD-10d VS SD-15d comparison, there were seven DEGs in the circadian rhythm pathway, including four up-regulated genes and three down-regulated genes (Fig. 5C).

Four light-related pathways were enriched in the SD-5d VS SD-15d comparison: antenna protein, circadian rhythm, plant hormone, and photosynthesis pathways, including 9, 17, 45, and 14 DEGs, respectively. The expression patterns of genes in different pathways differed. All nine DEGs in the antenna protein pathway were up-regulated. Of the 17 DEGs in the circadian rhythm pathway, three were up-regulated and

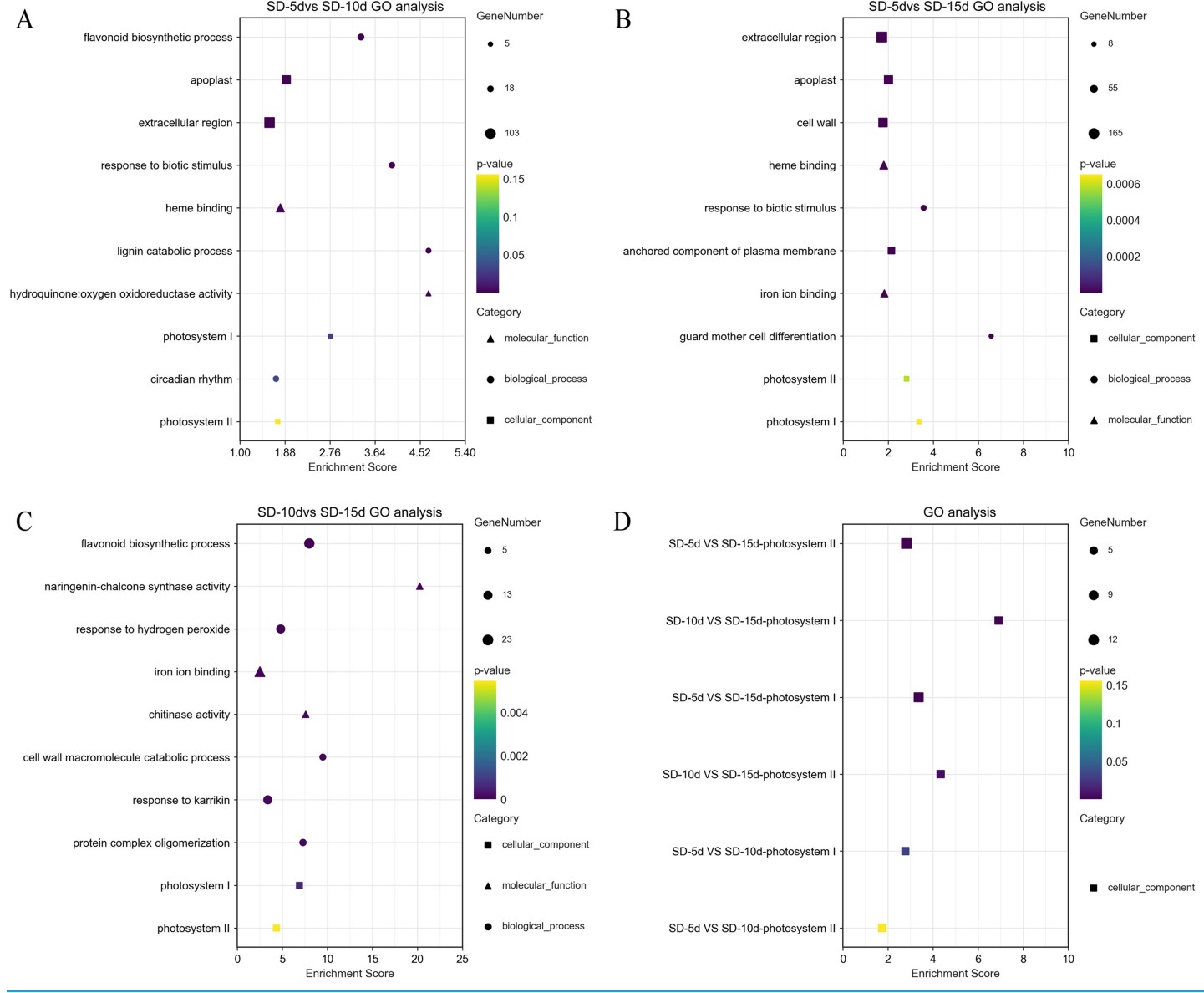

**Figure 4** **Results of GO enrichment in SD-5d VS SD-10d, SD-5d VS SD-15d, and SD-10d VS SD-15d three groups.** GO analysis of (A) SD-5d VS SD-10d, (B) SD-5d VS SD-15d and (C) SD-10d VS SD-15d comparison groups under short-day induction treatment. (D) The GO enrichment results entries were common to three comparison groups.

14 were down-regulated. A total of 45 DEGs were identified in the plant hormone signal transduction pathway, consisting of 27 up-regulated and 18 down-regulated genes. All 14 DEGs in the photosynthesis pathway were up-regulated (Fig. 5B). These results showed that all three comparison groups contained two pathways related to antenna proteins and circadian rhythms and that the SD-5d VS SD-10d and SD-5d VS SD-15d comparisons had the most significant degrees of enrichment in the circadian rhythm pathway (Fig. 5D). This suggested that different short-day induction periods, on the one hand, activated gene expression in the antenna protein pathway and promoted photosynthesis, and on the

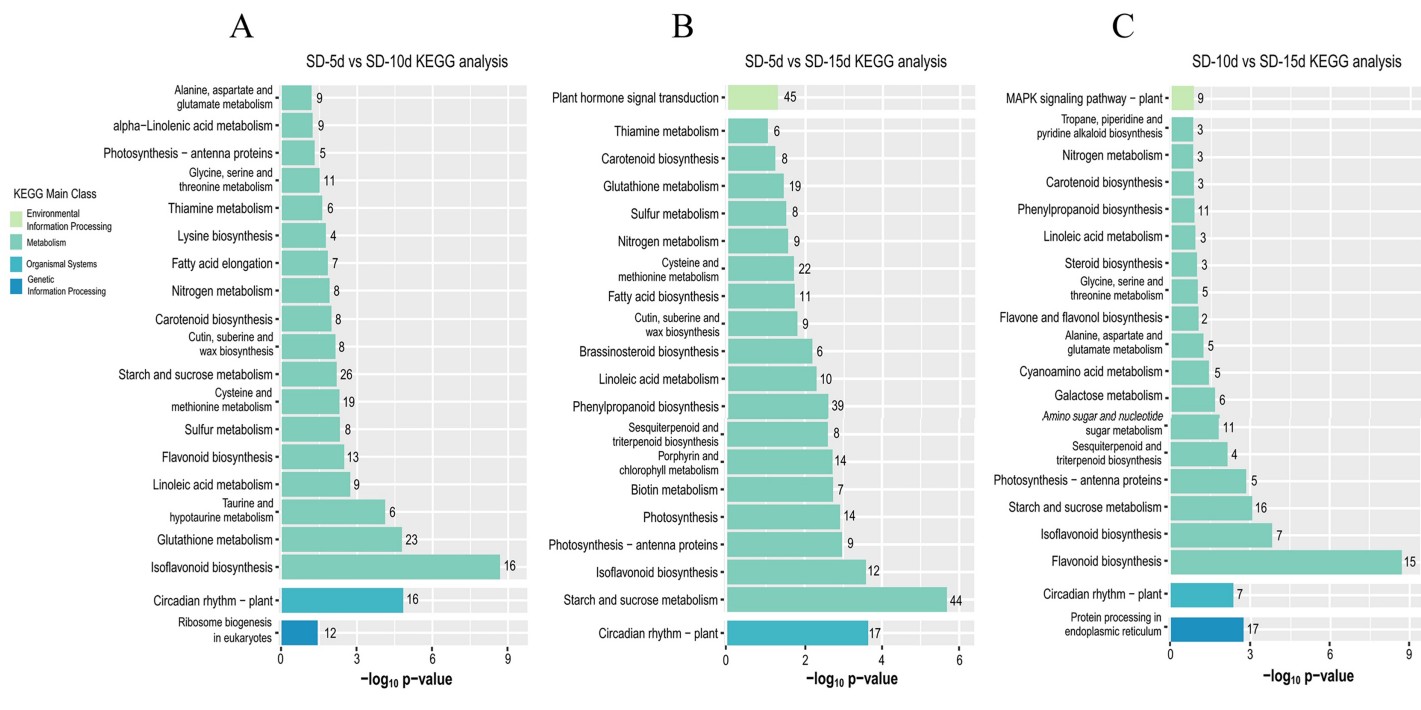

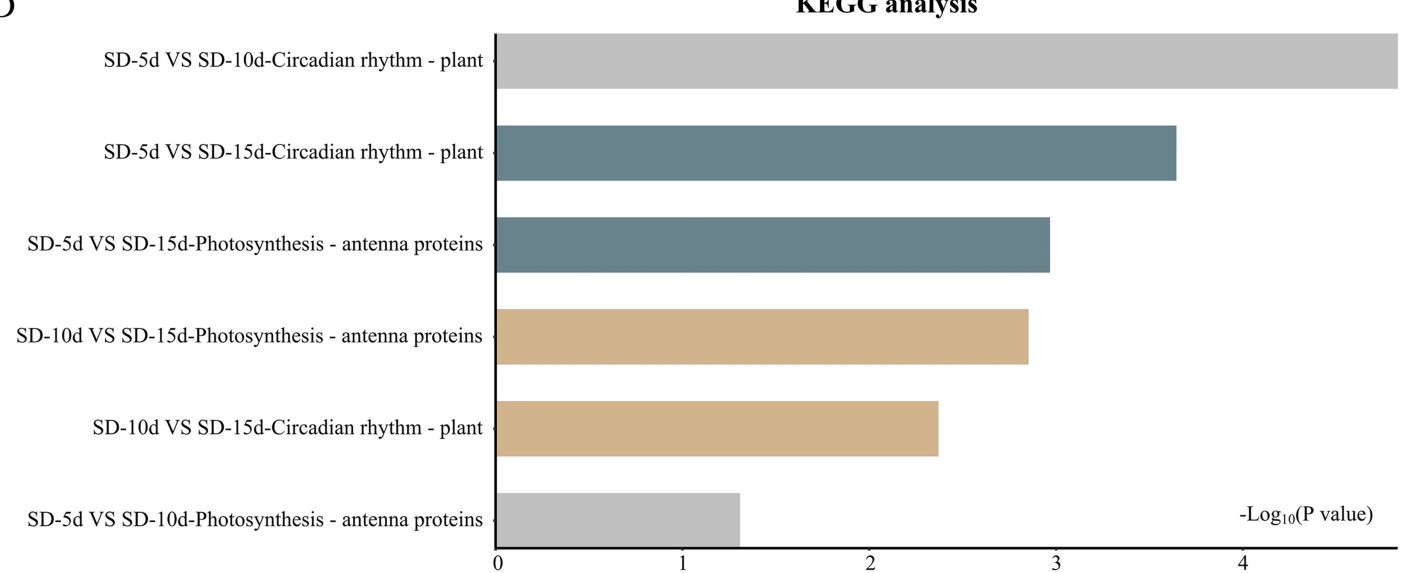

**Figure 5 Results of the KEGG enrichment in three group comparing SD-5d VS SD-10d, SD-5d VS SD-15d, and SD-10d VS SD-15d.** KEGG analysis of (A) SD-5d VS SD-10d, (B) SD-5d VS SD-15d and (C) SD-10d VS SD-15d comparison groups under short-day induction treatment. (D) The KEGG enrichment results are common to three comparison groups.

contrary, regulated photosynthesis, stomatal rhythmic opening and closing, leaf rhythmic opening and closing, and other important physiological processes, by regulating the biological clock. This may help adzuki bean plants to optimize their energy utilization efficiency and guarantee early flowering. Additionally, the circadian pathway played a key role in responding to different short-day induction periods.

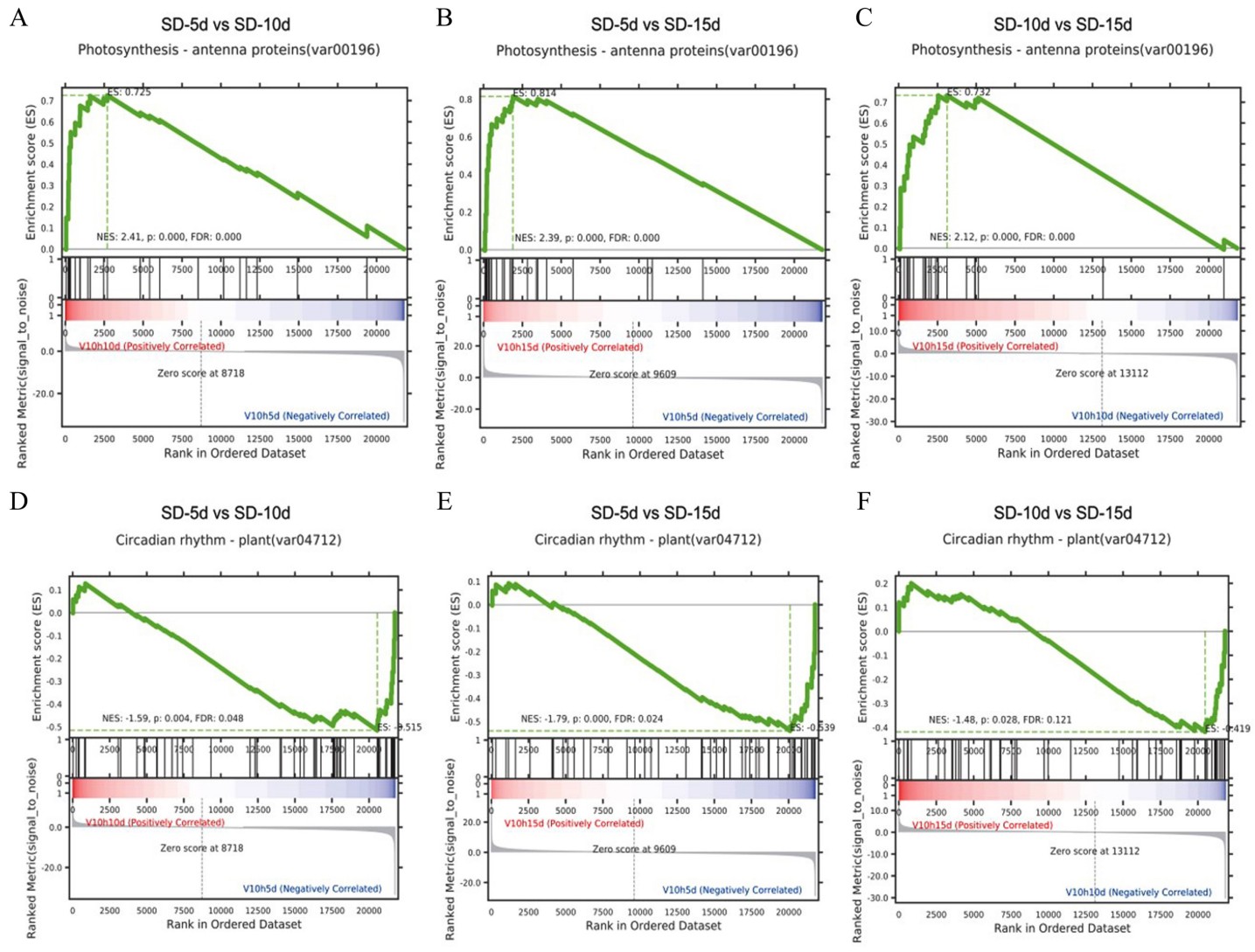

**Figure 6 Gene set enrichment analysis of antenna proteins and circadian rhythm pathway.** (A–C) Gene set enrichment analysis of antenna proteins in SD-5d VS SD-10d, SD-5d VS SD-15d and SD-10d VS SD-15d three comparison groups. (D–F) Gene set enrichment analysis of circadian rhythm pathway in SD-5d VS SD-10d, SD-5d VS SD-15d and SD-10d VS SD-15d three comparison groups.

## Gene set enrichment analysis in the KEGG pathway

It is generally believed that a positive enrichment score (ES) value indicates that a certain functional gene set is enriched at the front of the ranked sequence and that the involved pathway is up-regulated, and a negative ES value indicates that a certain functional gene set is enriched at the rear of the ranked sequence and that the involved pathway is down-regulated (*Subramanian et al., 2005*). The core genes that appeared in the ES map indicated that this functional gene set had biological significance. Here, gene set enrichment analysis revealed that the antenna proteins and circadian rhythm pathways in the three comparison groups had the same enrichment scores (Fig. 6). The antenna protein pathway was up-regulated (Figs. 6A–6C), the circadian rhythm pathway was down-regulated (Figs. 6D–6F), and core genes were identified in both pathways. The core

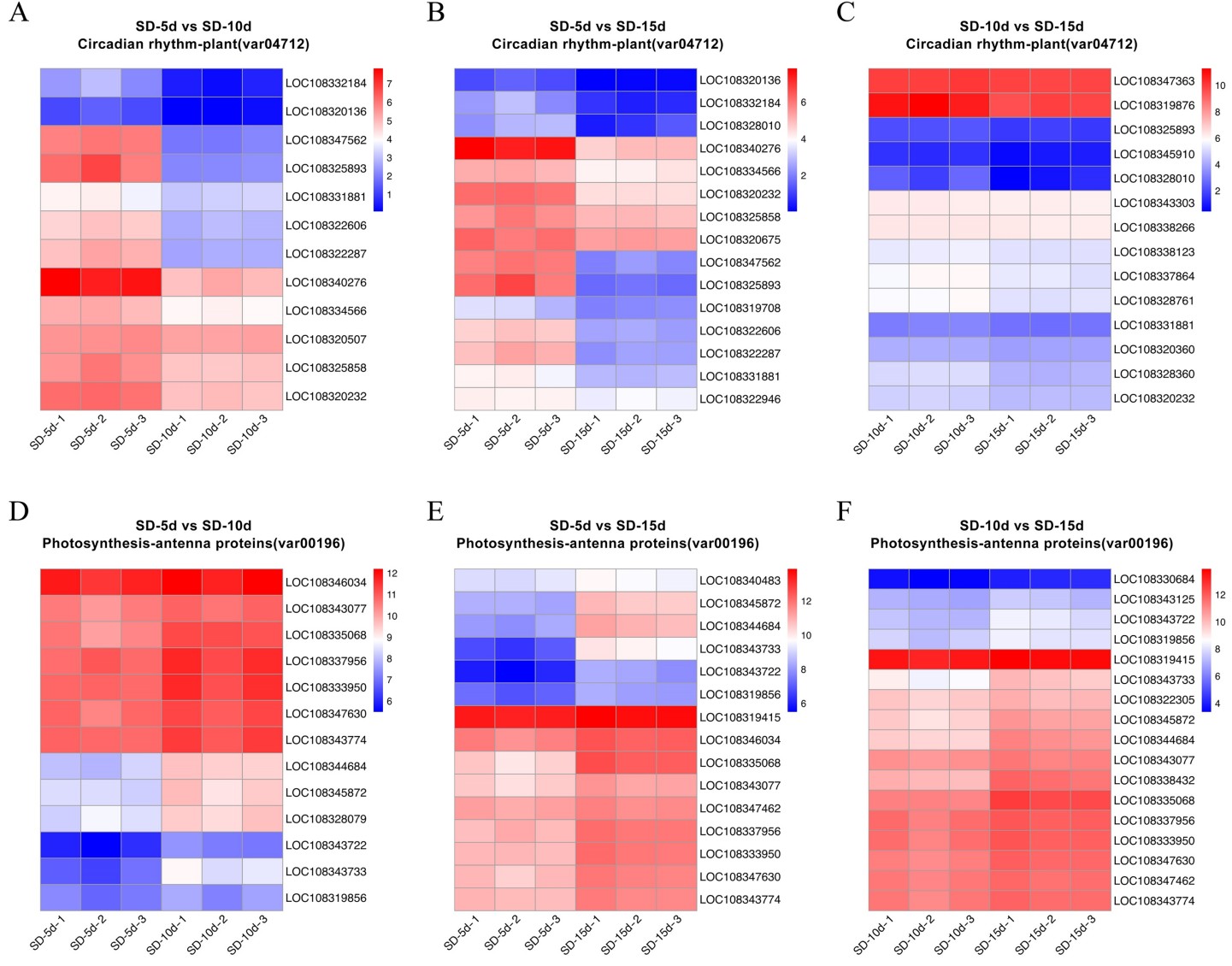

**Figure 7 Heatmap cluster analysis of differential genes in antenna proteins and circadian rhythm metabolic pathways.** (A–C) Enrichment analysis of circadian rhythm pathways in SD-5d VS SD-10d, SD-5d VS SD-15d and SD-10d VS SD-15d three comparison groups. (D–F) Enrichment analysis of antenna proteins pathways in SD-5d VS SD-10d, SD-5d VS SD-15d and SD-10d VS SD-15d three comparison groups.

genes of the antenna protein pathway were enriched in the front of the sequencing sequence; whereas, those of the circadian rhythm pathway were enriched at the rear of the sequencing sequence in the three comparison groups, indicating that these core genes played important roles in the response of adzuki bean to short-day induction.

Heatmap cluster analysis was performed on the core genes of antenna protein and circadian rhythm metabolic pathways (Fig. 7). The circadian rhythm pathways in the SD-5d VS SD-10d, SD-5d VS SD-15d, and SD-10d VS SD-15d comparisons contained 12, 15, and 14 core genes, respectively, which were inversely down-regulated as the number of short-day induction days increased (Figs. 7A–7C). The antenna protein pathways of the three comparisons, contained 13, 15, and 17 core genes, respectively, which were

up-regulated as the number of short-day induction days increased (Figs. 7D–7F). These results indicated that adzuki bean responded to different short-day induction periods by up-regulating key genes in the antenna protein pathway or down-regulating key genes in the circadian rhythm pathway, thus promoting early flowering.

## Analysis of circadian and antenna protein pathway maps

The circadian rhythm involves a rhythmic change in plants in response to daylight length. Analysis of the circadian rhythm metabolic pathway showed that blue light promoted flowering and far-red light inhibited flowering in adzuki bean. After short-day induction, *PHYB* (*i.e.*, a far-red light receptor interacted with *PIF3*) (*i.e.*, a phytochrome interaction factor) to promote NDA transcription and production of *LHY* (*i.e.*, a circadian clock regulatory protein). The PRR7 protein of the central oscillator-controlled internal clock inhibited *PIF3* transcription and *LHY* production. In addition, *PRR7* (*i.e.*, a key gene in the photoperiodic flowering pathway) inhibited LHY protein expression, thus, inhibiting adzuki bean flowering. After short-day induction, one pathway regulating flowering involved the inhibition of complex formation between the morphogenesis regulator *COP1* and the signal sensing protein SPA1 by the cryptochrome CRY-blue light receptor protein. This complex was down-regulated and ubiquitinated to inhibit transcription of *HY5* (*i.e.*, a negative regulator of photomorphogenesis), thus indirectly promoting photomorphogenesis. Another pathway regulating flowering was the CRY pathway. CRP is a blue light receptor protein that inhibited formation of a complex between *COP1* and *ELF3*, thus inhibiting *GI* by ubiquitylation. As a negative regulator, *GI* down-regulated the function of the complex formed by *FKF1* and *GI*; inhibited *CDF1* binding to the CO promoter by ubiquitylation; and thus, initiated transcriptional activation of *CO* and up-regulation of *FT*. *GI* also directly promoted *CO* transcription, which can up-regulate *FT*, and indirectly promoted adzuki bean flowering (Fig. 8A). Figure 8B shows that changes in the expression levels of seven protein-coding genes (*i.e.*, *PIF3*, *PRR7*, *SPA1*, *HY5*, *ELF3*, *GI*, and *FKF1*) differed among the SD-5d VS SD-10d, SD-5d VS SD-15d, and SD-10d VS SD-15d comparisons. More significant changes were observed in the SD-5d VS SD-15d comparison than in other comparisons, indicating that a longer short-day induction period caused greater gene expression change and that most genes were down-regulated.

Chloroplasts in green plants are comprised of PSI and PSII (*Ferroni et al., 2016*). PSI contains five light-trapping pigment protein complexes (*i.e.*, Lhca1, Lhca2, Lhca3, Lhca4, and Lhca5) and PSII contains six (*i.e.*, Lhcb1, Lhcb2, Lhcb3, Lhcb4, Lhcb5, and Lhcb6), of which Lhcb1–3 play major roles (*Green, Pichersky & Kloppstech, 1991*). Different short-day induction periods may affect the expression of genes encoding antenna proteins, allowing them to adapt to adzuki bean growth and development. Figures 9A and 9B show that the genes encoding the Lhca3, Lhcb1, Lhcb2, Lhcb3, Lhcb4, and Lhcb6 proteins were all up-regulated after a short-day induction period, and that there was a greater number of up-regulated genes encoding the Lhcb1, Lhcb2, and Lhcb3 proteins than of those encoding the other proteins. Figure 9B shows that the largest number of DEGs occurred in the SD-5d VS SD-15d comparison, indicating that the light signal reception ability was enhanced with an increase in the number of short-day induction days; that is, a longer short-day

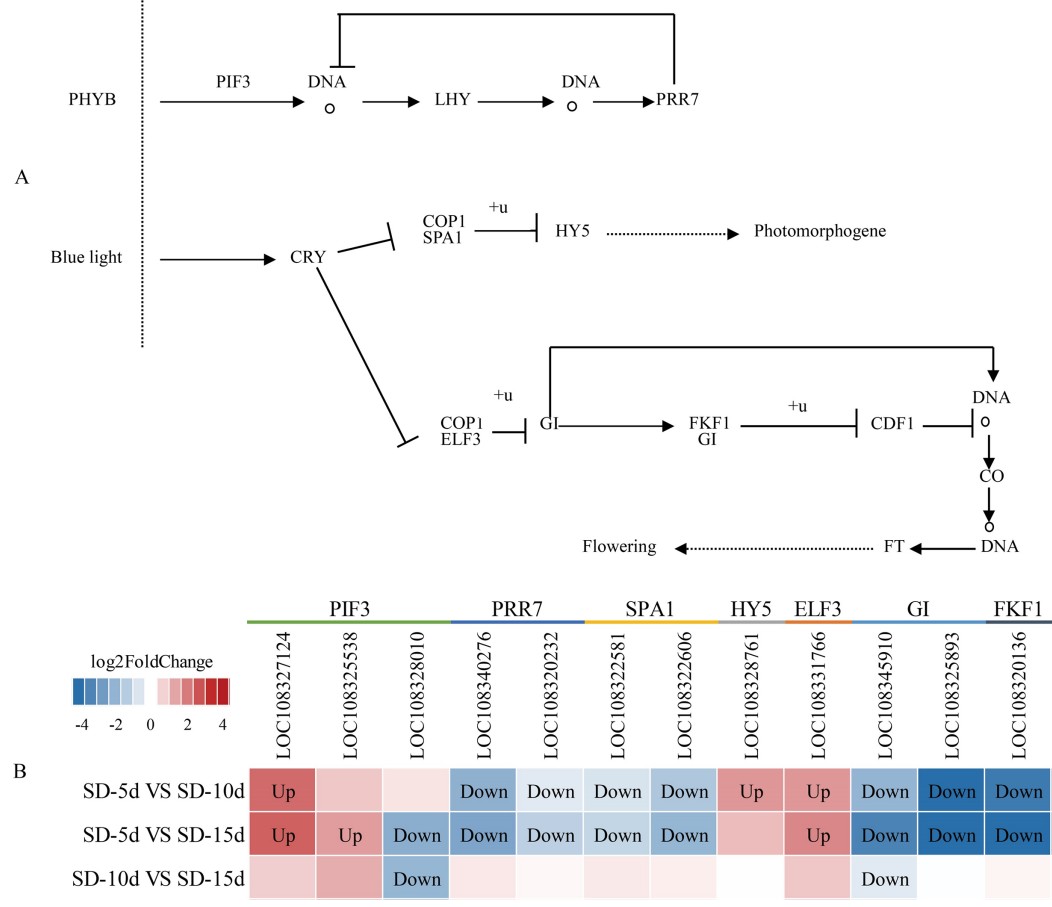

**Figure 8 Circadian rhythm metabolic pathway and analysis of gene expression in this pathway.**
(A) Circadian rhythm metabolic pathway of adzuki bean. (B) Analysis of gene expression in SD-5d VS SD-10d, SD-5d VS SD-15d and SD-10d VS SD-15d three comparison groups of circadian rhythm metabolic pathway.               

induction period resulted in stronger reception ability. These results indicated that adzuki bean responded to short-day induction by up-regulating PSI- and PSII-related genes. These genes may play key roles in regulating adzuki bean seedling growth and development under short-day induction conditions, and a longer induction period resulted in greater changes in gene expression levels.

## Identification of DEGs by qRT–PCR

To verify the reliability and accuracy of the transcriptomic data, samples from the same treatment group were subjected to qRT‒PCR analysis. Eight genes with significant variability were screened from the circadian rhythm and antenna protein pathways, all of which were closely related to light-related functions and associated with flowering based on homologous sequence alignment (Table 4). qRT‒PCR was performed to verify the differential expression levels of these eight genes in SD-5d VS SD-10d, SD-5d VS SD-15d, and SD-10d VS SD-15d comparisons.

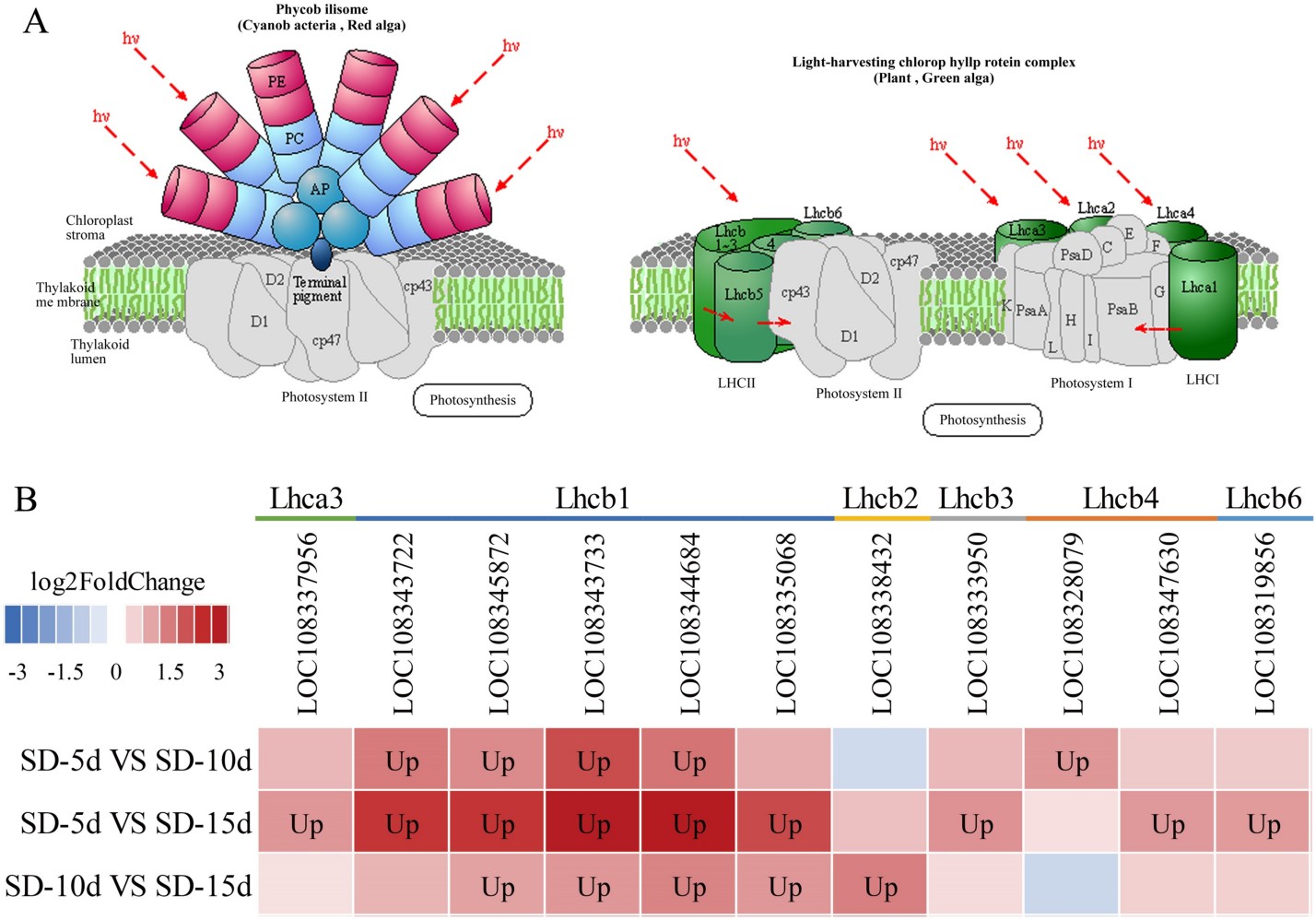

**Figure 9 Antenna proteins metabolic pathway and analysis of gene expression in this pathway.** (A) Antenna proteins metabolic pathway diagram of adzuki bean. (B) Analysis of gene expression in SD-5d VS SD-10d, SD-5d VS SD-15d and SD-10d VS SD-15d three comparison groups of antenna proteins metabolic pathway.

The *LOC108331766* gene expression levels were significantly different in the SD-5d VS SD-10d and SD-5d VS SD-15d comparisons, but not in the SD-10d VS SD-15d comparison. The *LOC108322606* gene showed significantly different expression levels in the SD-5d VS SD-10d and SD-10d VS SD-15d comparisons but not in the SD-5d VS SD-15d comparison. In the antenna protein pathway, *LOC108345872* and *LOC108344684* genes showed significantly different expression levels among the three groups. The *LOC108335068* gene showed significantly different expression levels in the SD-5d VS SD-15d and SD-10d VS SD-15d comparisons, but not in the SD-5d VS SD-10d comparison. *LOC108328079*, *LOC108333950*, and *LOC108338432* genes showed significantly different expression levels in the SD-5d VS SD-10d, SD-5d VS SD-15d, and SD-10d VS SD-15d comparisons, respectively, but not in the other comparisons. Therefore, the qRT-PCR results for these eight genes were consistent with the RNA-seq results (Figs. 10A1B1, 10A2B2, 10A3B3, and 10A4B4). These eight genes in the circadian

**Table 4 List of different genes associated with light in circadian rhythm and antenna proteins signaling pathways.**

| Gene ID | LOC108331766 | LOC108322606 | LOC108345872 | LOC108328079 | LOC108344684 | LOC108335068 | LOC108333950 | LOC108338432 |
|---|---|---|---|---|---|---|---|---|
| KEGG map | Circadian rhythm-plant (KEGG map) | | Photosynthesis-antenna proteins (KEGG map) | | | | | |
| Gene symbol | F17H15.25 | F11C10.3 /F11C10.4 | CAB21 | F27I1.2 | CAB1B | CAB3 | LHBC1 | LHCB2 |
| Regulation | Up | Down | Up | Up | Up | Up | Up | Up |
| Location | Chromosome 4 NC_030640.1 | Chromosome Un NW_016115133.1 | Chromosome 10 NC_030646.1 | Chromosome 3 NC_030639.1 | Chromosome 10 NC_030646.1 | Chromosome 6 NC_030642.1 | Chromosome 5 NC_030641.1 | Chromosome 7 NC_030643.1 |
| Description | Protein early flowering 3-like | Protein suppressor of phya-1051-like | Chlorophyll a-b binding protein of LHCII type 1-like | Chlorophyll a-b binding protein CP29.3, chloroplastic | Chlorophyll a-b binding protein of LHCII type 1-like | Chlorophyll a-b binding protein of LHCII type 1-like | Chlorophyll a-b binding protein 13, chloroplastic-like | Chlorophyll a-b binding protein 215, chloroplastic |
| Differential groups | SD-5d VS SD-0d; SD-5d VS SD-15d | SD-5d VS SD-0d; SD-10d VS SD-15d | SD-5d VS SD-d; SD-5d VS SD-15d; SD-10d VS SD-15d | SD-5d VS SD-d | SD-5d VS SD-0d; SD-5d VS SD-15d; SD-10d VS SD-15d | SD-5d VS SD-5d; SD-10d VS SD-15d | SD-5d VS SD-15d | SD-10d VS SD-5d |
| q-value | 2.83E-03 6.78E-06 | 5.84E-33 1.44E-51 | 3.64E-08 4.27E-39 1.23E-05 | 8.03E-07 | 6.62E-14 1.19E-41 3.09E-14 | 4.91E-18 1.58E-12 | 2.32E-12 | 6.97E-15 |
| Functional annotations of orthologs | Protein early flowering 3-like | WD40 repeat | Chlorophyll A-B binding protein | Chlorophyll A-B binding protein | Chlorophyll A-B binding protein | Chlorophyll A-B binding protein | Chlorophyll A-B binding protein | Chlorophyll A-B binding protein |

rhythm and antenna protein pathways showed significantly different expression levels, indicating that the transcriptome sequencing data were accurate and reliable.

# DISCUSSION

The life cycle of higher plants includes vegetative and reproductive growth. Flower initiation marks the transition between plant growth stages and determines whether a plant can successfully reproduce (*Blümel, Dally & Jung, 2015*). As an important agronomic index, flowering time is a prerequisite for high yield and a high harvest index of crops and it is greatly affected by the photoperiod (*Kinoshita & Richter, 2020*). In this study, a short-day sensitive variety, 'Tangshan hong xiao dou,' was used under short-day treatments for 5, 10, and 15 d. A longer short-day induction period initiated earlier flowering and had a greater inhibitory effect on adzuki bean. These results indicated that adzuki bean promotes flowering by inhibiting plant growth after short-day induction, which is consistent with the results of studies on chrysanthemums and soybeans (*Lu & Yang, 2021*; *Yang et al., 2021a*).

The plant light-dependent signaling response is an extremely complex process involving the regulation of many related genes. To further explore the regulatory mechanism of short-day induction of adzuki bean flowering and expression of key genes, a cDNA library of adzuki bean leaves was constructed after different short-day induction periods and transcriptome sequencing was conducted using Illumina sequencing technology. A total of

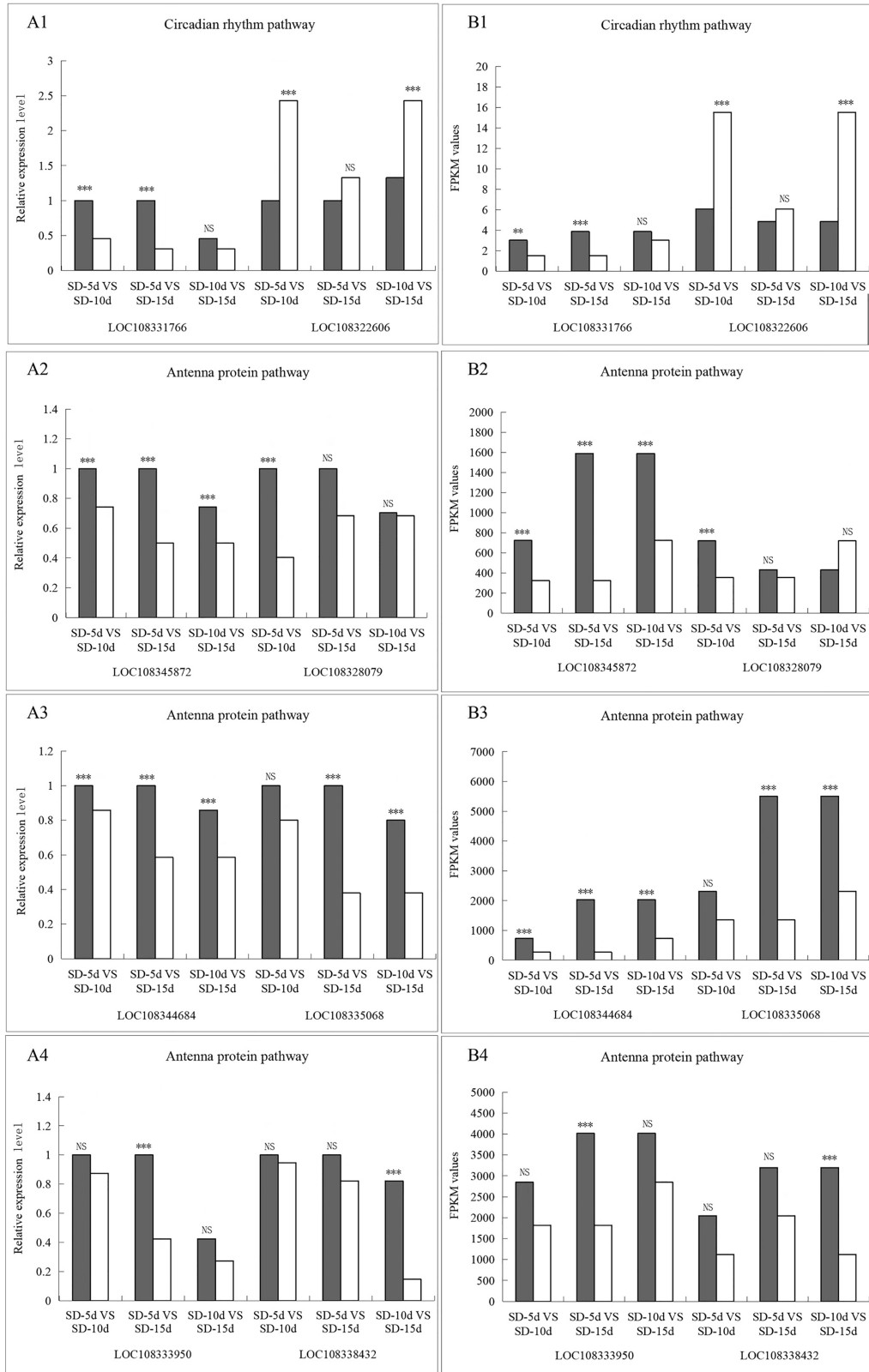

**Figure 10 The qRT-PCR validation of eight differentially related photoperiod expressed genes in circadian rhythm and antenna proteins pathways.** (A1B1) The qRT-PCR validation of two genes in

**Figure 10** (continued)
the circadian pathway under SD-5d VS SD-10d, SD-5d VS SD-15d and SD-10d VS SD-15d three comparison groups. (A2B2, A3B3, A4B4) The qRT-PCR validation of six genes in the antenna protein pathway under SD-5d VS SD-10d, SD-5d VS SD-15d and SD-10d VS SD-15d three comparison groups. Asterisks (**) indicate significance at the 0.01 probability levels, (***) indicate significance at the 0.001 probability levels, respectively; NS, not significant.     

5,939 DEGs were obtained in the SD-5d VS SD-10d, SD-5d VS SD-15d, and SD-10d VS SD-15d comparisons. The SD-5d VS SD-15d comparison showed the most DEGs (3,068). GO functional enrichment results showed that PSI and PSII, which have two light-related functions, were both enriched in the DEGs from the three comparisons, and the number of DEGs identified in each comparison differed. There were nine and 12 genes corresponding to PSI and PSII, respectively, in the SD-5d VS SD-15d comparison. All nine PSI-associated genes were up-regulated; whereas, among the 12 PSII-associated genes, 11 were up-regulated and one was down-regulated. These findings indicated that a longer short-day induction period resulted in a higher number of DEGs, that most of these genes were up-regulated, and that more pathways were enriched. The genes in these pathways interacted to regulate the adzuki bean flowering time. PSI and PSII are related to plant light-trapping pigment protein complexes that receive solar energy and assimilate $CO_2$ to form chemical energy (*De Bianchi et al., 2011*). There are three main light-trapping pigment protein complexes encoded by Lhcb1, Lhcb2, and Lhcb3 in the peripheral light-trapping antenna of PSII, the main function of this complex is to capture and transfer light energy and balance the excitation energies of PSI and PSII, it also plays an important role in maintaining the membrane structure of thylakoids in response to changes in the external environment and photoprotection (*Xia et al., 2012*; *Xu et al., 2012*). From these results, it can be speculated that the photosynthetic capacity of adzuki bean was enhanced by up-regulation of PSI- and PSII-related genes after short-day stress. In particular, the photocapturing capacity of the peripheral photocapture antenna of PSII was enhanced. In addition, the electron transfer rate was accelerated to rapidly respond to changes in the external environment and maintain normal growth and development. The subsequent gene pathway map of antenna protein gene expression also showed that genes in the three comparisons were up-regulated, and the number of DEGs was the highest in the SD-5d VS SD-15d comparison. Among these, five genes encoding the Lhcb1 protein were up-regulated, suggesting that this protein played an important role in light energy absorption after short-day induction in adzuki bean.

There is a complex interaction between exogenous photoperiodic signals and the flowering biological clocks of plants when sensing different photoperiod changes. This interaction causes a series of activations or inhibitions of flowering genes, ultimately leading to changes in the flowering rhythm (*Song, Ito & Imaizumi, 2010*). Different plant flowering rhythms and photoperiodic pathways are largely conserved (*Wang et al., 2019*). In this study, KEGG enrichment analysis showed that two light-related antenna protein and circadian rhythm pathways were enriched in three comparisons, and that the enrichment degree of the circadian rhythm pathways was most significant in the SD-5d VS SD-15d comparison group. Further gene set enrichment analysis revealed that adzuki bean

responded to short-day induction by up-regulating the antenna protein pathway and down-regulating key genes in the circadian rhythm pathway, thus promoting early flowering. This is similar to the findings of previous studies on different species of edible fungi and peaches under short-day induction using transcriptome sequencing (*Xiao et al., 2017*; *He et al., 2017*). Further analysis of the regulatory network showed that the antenna protein enhanced light energy absorption through gene up-regulation in the circadian rhythm pathway. Early flowering was promoted through the blue light metabolic pathway, whereas far-red light inhibited adzuki bean flowering. Early studies on *A. thaliana* found that the red-light receptor *PHYB* and the blue light receptor *CRY2* have antagonistic effects on flowering time regulation (*Blackman et al., 2010*), and that *CRY1*, *CRY2*, and *PHYA* stabilize *CO*, whereas *PHYB* promotes its degradation (*Ishikawa, Kiba & Chua, 2006*).

How does blue light regulate the CO protein in adzuki bean? Our study found that one regulatory pathway is the far-red light signaling pathway in the circadian rhythm system. In this pathway, *PHYB* binds to *PIF3* to promote NDA transcription to produce *LHY*, and *PRR7* inhibits *PIF3* transcription to produce *LHY* protein, thus inhibiting flowering. Another pathway is the blue light pathway, in which *CRY* inhibits formation of the COP1/SPA1 complex, and its down-regulation inhibits *HY5* transcription through ubiquitination, thus indirectly promoting light morphogenesis. In addition, *CRY* inhibited formation of the COP1/ELF3 complex and *GI* through ubiquitination. As a negative regulator, *GI* down-regulated formation of the FKF1/GI complex, inhibited *CDF1* binding to the *CO* promoter through ubiquitination, transcriptionally activated *CO*, and up-regulated *FT* to indirectly promote adzuki bean flowering. *GI* also directly promotes *CO* transcription, which in turn promotes *FT* up-regulation, thereby indirectly promoting adzuki bean flowering. Based on above analysis, light regulated *CO* activity through at least two processes: the biological clock regulated *CO* mRNA expression and stability of the *CO* protein was regulated through signal transduction of different light receptors.

In *A. thaliana*, *CRY2* cannot interact with *SPA1* during the dark period, when high *CO* gene expression levels occur under short-day conditions, which leads to the degradation of the *CO* protein by the COP1/SPA1 complex and subsequently, flowering (*Zuo et al., 2011*). Under long-day conditions, GI and FKF1 proteins, when present at high levels, form a complex and degrade the CDF1 protein under blue light, which eliminates the inhibitory effect of *CDF1* on the *CO* gene, subsequently, CO mRNA begins to accumulate, promoting up-regulation of *FT* genes and flowering (*Jang et al., 2008*). The flowering regulation mechanism of adzuki bean was similar to that of *A. thaliana*. Substantial progress has been made in the study of cryptochrome genes in soybean (*Xia et al., 2022*; *Liu et al., 2022*; *Lin et al., 2021*), *A. thaliana* (*Chen et al., 2021*; *Yan et al., 2020*), and rice (*Xu et al., 2020*). Cryptochrome acts as a photoreceptor in various angiosperms, regulates the biological clock, and plays an important role in induction of flowering. However, the number of up-regulated or down-regulated genes varies between flowering plants. Eight DEGs identified in circadian rhythm and antenna protein pathways in the present study were consistent between the RNA-seq and qRT−PCR results, the functions of which were related to light and flowering. This study clarified the regulatory mechanism of short-day induction of adzuki bean flowering. The number of days required for flowering and

maturation of different adzuki bean varieties under different short-day induction periods can be broadly determined through the abovementioned studies. This provides a practical reference for introducing adzuki bean into different regions. When breeding different adzuki bean varieties, it may be possible to synchronize flowering through short-day induction and accelerate the breeding process.

## CONCLUSIONS

The number of DEGs in the SD-5d VS SD-15d comparison was 3,068. A longer short-day induction time resulted in greater promotion of flowering. This regulatory mechanism was associated with down-regulation of key genes in the circadian rhythm pathway, thus regulating the blue light metabolism pathway to promote flowering under short-day induction. In addition, the antenna protein pathway accelerated electron transfer through gene up-regulation to promote adzuki bean flowering. Eight DEGs screened from two metabolic pathways were mostly up-regulated and were verified as candidate genes for regulating the adzuki bean flowering time using RNA-seq. These results provide valuable information for the design of functional studies of flowering-related genes and lay the foundation for controlling the adzuki bean flowering time by regulating gene expression through gene overexpression, knockdown, or knockout. Synchronization of flowering in different adzuki bean varieties may accelerate the breeding process. In addition, adzuki bean varieties with high shade tolerance may be selected by comparing different short-day induction responses. These data can be used to determine suitable intercropping crop varieties for planting under forests.

### Funding
This work was supported by the Hebei Province Natural Science Foundation (C2021204045) and the Hebei Agriculture Research System (HBCT2024070203). The funders had no role in study design, data collection and analysis, decision to publish, or preparation of the manuscript.

### Grant Disclosures
The following grant information was disclosed by the authors:
Hebei Province Natural Science Foundation: C2021204045.
Hebei Agriculture Research System: HBCT2024070203.

### Competing Interests
The authors declare that they have no competing interests.

### Author Contributions
- Weixin Dong conceived and designed the experiments, analyzed the data, prepared figures and/or tables, authored or reviewed drafts of the article, and approved the final draft.

- Dongxiao Li analyzed the data, authored or reviewed drafts of the article, and approved the final draft.
- Lei Zhang performed the experiments, analyzed the data, prepared figures and/or tables, authored or reviewed drafts of the article, and approved the final draft.
- Peijun Tao performed the experiments, prepared figures and/or tables, authored or reviewed drafts of the article, and approved the final draft.
- Yuechen Zhang conceived and designed the experiments, prepared figures and/or tables, authored or reviewed drafts of the article, and approved the final draft.

## Data Availability

The *Vigna angularis* sequences are available at NCBI: PRJNA817421.

## Supplemental Information

Supplemental information for this article can be found online at http://dx.doi.org/10.7717/peerj.17716#supplemental-information.

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
