# Peer review of "Flowering-associated gene expression and metabolic characteristics in adzuki bean (Vigna angularis L.) with different short-day induction periods"

_PeerJ, doi:10.7717/peerj.17716_

## Round 0.1 · original submission · Major Revisions

Following the thoughtful feedback and suggestions provided by our esteemed reviewers, I have decided that the manuscript requires major revisions before further consideration. The authors must address each point raised by the reviewers thoroughly and provide clear justifications or modifications in response. The required changes are significant and should be elucidated point by point in the revised submission. This approach will ensure the concerns are adequately addressed and will enhance the clarity, accuracy, and contribution of the manuscript to our field.

·

Basic reporting

The introduction is too long with a lot of details.

Experimental design

No comment

Validity of the findings

No comment

Reviewer 2 ·

Basic reporting

The manuscript titled “Analysis of flowering-associated gene expressions and metabolic characteristics in adzuki bean (Vigna angularis L.) with different short-day induction” discusses the effects of different day-length on the expression and metabolic characteristics of genes related to flowering time in adzuki beans. The study conducted field experiments using three treatments with varying shading durations. Transcriptome sequencing was performed, and the results showed differentially expressed genes (DEGs) related to photosystem I and II. The study also identified two metabolic pathways, the circadian rhythm pathway and the antenna protein pathway which play a role in promoting flowering in adzuki beans. Real-time qRT-PCR validation of DEGs confirmed the accuracy of the sequencing results. The findings suggest that short-day induction can down-regulate genes in the circadian rhythm pathway and upregulate genes in the antenna protein pathway, providing insights into the molecular mechanisms of adzuki bean flowering induced by short days. The study contributes to the understanding of key genes regulating adzuki bean flowering. The manuscript is written in a clear and logical way but there are still a few points that need clarification or further details.

Experimental design

Well designed, many details have been provided, such as the soil nutrients content or the filed microclimate changes.

Validity of the findings

1. Suggest to visualize Table 4 which shows different plant height, stem diameter and leaf area of adzuki bean under different short-day inducement times to bar plots or box plots.
2. Figure 1, the author should describe what specific test was done for statistics in the legend. Please add the downward error bar as well.
3. It’s better to add the PCA or NMDS analysis showing the reproducibility of all the samples.
4. For figure 1A and 1B, please add scale bar or ruler taken within the photos.
5. Figure 2A, it’s better to use the Volcano Plot.
6. Figure 2E, it’s better to generate the heatmap using all 3 replicates.
7. Please enhance the resolution of figure 8.
8. Figure 9, please keep the title of each plot consistent. It’s better to add NS for the samples without significant changes. Please add statistical details in the legend.
9. Figure 6, it’s better to generate the heat-map using all 3 replicates.

Additional comments

1. Line 19, I suggest simplifying and refining the abstract, particularly the Methods and Results sections.
2. The format of in-text citations should follow the instructions: https://peerj.com/about/author-instructions/#reference-format . Please also check the alignment of each Reference.
3. Line 51, I suggest “so the yield of adzuki beans is low and unstable” to be “resulting in low and unstable yield of adzuki beans”.
4. Line 52, suggest “sunshine duration” to be “photoperiod”.
5. Line 65, Arabidoopsis thaliana should be A. thaliana.
6. Line 67&68, need citation.
7. Line 67, what is CO and FT protein, COP1/SPA1?
8. Line 86, what TOC1 is has not been mentioned in previous text.
9. Line 113, suggest to rewrite “earlier than previously” for improved clarity.
10. Line 127, suggest “two-rows planting” to be planting in two rows.
11. Line 128, treatments should be treatment.
12. Line 152, plant should be Plant.
13. Line178, please clarify what method or kit was used for RNA extraction.
14. Figure 3 caption, please rewrite “three group comparing” and legend “short-day induction treatment different days” for an improved grammar and clarity.

Reviewer 3 ·

Basic reporting

The manuscript gives a concise overview of the history of adzuki beans and the need to comprehend how short-day inductions affect blooming mechanisms. It covers background, techniques, results, and debate and is logically organized from introduction to conclusion. The study's justification is well supported by the necessary references; however, a more comprehensive comparison with other crops and model organisms would enhance the literature evaluation. While some figures and tables could be improved for impact and clarity, they are still relevant and aid in understanding the study's conclusions.

Experimental design

Strong research objectives and an extensive technique including fieldwork, RNA sequencing, and qRT-PCR gene expression validation characterize the experimental design. Methodological rigor is demonstrated by the use of a randomized block design for field experiments and thorough protocols for RNA extraction and sequencing. While there is enough detail in the methods section to ensure reproducibility, there are some statistical analysis details that could be expanded upon to improve transparency.

Validity of the findings

Appropriate analyses, such as KEGG pathway enrichment, GO, and differential gene expression analysis, are reported with the data. The study provides fresh insights into the genetic regulation of flowering time in response to short-day circumstances by identifying particular metabolic pathways involved in adzuki bean flowering. The data provide strong support for the conclusions, but a more thorough comparison with previous research would help put the findings in a larger context and enhance the debate. To increase the validity of the conclusions, further information on the tests and statistical significance thresholds utilized would be beneficial.

Additional comments

With its fresh insights into the flowering mechanisms of adzuki beans under short-day conditions, this manuscript is a significant contribution to the field. The results hold significance for both fundamental plant biology and agricultural operations, and the study is well-designed. Nonetheless, the impact of the work might be increased with revisions to its organization, clarity, and scope of the literature review. More specifically, it would be helpful to streamline intricate justifications and broaden the conversation to encompass more extensive parallels with another research. The manuscript would also benefit from improved clarity in graphics and tables as well as more specific information about statistical analysis.

Annotated reviews are not available for download in order to protect the identity of reviewers who chose to remain anonymous.

---

## Round 0.2 · Minor Revisions

The authors have successfully implemented all the recommended revisions by the esteemed reviewer, resulting in a substantial enhancement of the manuscript's quality. However, the manuscript requires significant language proofreading to improve readability. I suggest the authors consult a fluent English speaker or a professional proofreading service to make the necessary changes before final Acceptance.

·

Basic reporting

no comment

Experimental design

no comment

Validity of the findings

no comment

Reviewer 2 ·

Basic reporting

The authors improved the quality of figures and revised the manuscript text for better clarity. Morevover, they provided more detailed explanations of their methods, and successfully incorporated the point to point wise through the manuscript as suggested comments so that I would like to strongly recommend that acceptance of manuscript in PeerJ for publication.

Experimental design

NA

Validity of the findings

NA

Additional comments

NA

---

## Round 0.3 · Minor Revisions

The manuscript still contains several issues related to language clarity in many sections. I recommend that the author sends the manuscript to a reputable proofreading service provider and obtains a certificate from them to verify the proofreading.

---

## Round 0.4 · accepted · Accept

I confirm that the authors have addressed all of the reviewers' comments thoroughly. I have personally assessed the revised version. I am pleased to report that I am satisfied with the revisions and the current version of the manuscript.